# Discrimination and Identification of Aroma Profiles and Characterized Odorants in Citrus Blend Black Tea with Different Citrus Species

**DOI:** 10.3390/molecules25184208

**Published:** 2020-09-14

**Authors:** Jiatong Wang, Yin Zhu, Jiang Shi, Han Yan, Mengqi Wang, Wanjun Ma, Yue Zhang, Qunhua Peng, Yuqiong Chen, Zhi Lin

**Affiliations:** 1Key Laboratory of Tea Biology and Resources Utilization, Ministry of Agriculture, Tea Research Institute, Chinese Academy of Agricultural Sciences, No. 9 Meiling South Road, Hangzhou 310000, China; wangjiatong@tricaas.com (J.W.); zhuy_scu@tricaas.com (Y.Z.); yanhan@tricaas.com (H.Y.); lfwangmengqife@126.com (M.W.); mawanjun@tricaas.com (W.M.); zhangyue@tricaas.com (Y.Z.); pqh@tricaas.com (Q.P.); 2Graduate School of Chinese Academy of Agricultural Sciences, No. 12 Zhongguancun South Street, Haidian District, Beijing 100081, China; 3College of Horticulture and Forestry Science, Huazhong Agricultural University, No. 1 Shizishan Street, Hongshan District, Wuhan 430000, China; chenyq@mail.hzau.edu.cn

**Keywords:** aroma profiles, characterized odorants, citrus blend black tea, discrimination, GC×GC-TOFMS, GC-O/MS

## Abstract

Citrus blend black teas are popular worldwide, due to its unique flavor and remarkable health benefits. However, the aroma characteristics, aroma profiles and key odorants of it remain to be distinguished and cognized. In this study, the aroma profiles of 12 representative samples with three different cultivars including citrus (*Citrus reticulata*), bergamot (*Citrus bergamia*), and lemon (*Citrus limon*) were determined by a novel approach combined head space-solid phase microextraction (HS-SPME) with comprehensive two-dimensional gas chromatography-time-of-flight mass spectrometry (GC×GC-TOFMS). A total of 348 volatile compounds, among which comprised esters (60), alkenes (55), aldehydes (45), ketones (45), alcohols (37), aromatic hydrocarbons (20), and some others were ultimately identified. The further partial least squares discrimination analysis (PLS-DA) certified obvious differences existed among the three groups with a screening result of 30 significant differential key volatile compounds. A total of 61 aroma-active compounds that mostly presented green, fresh, fruity, and sweet odors were determined in three groups with gas chromatography-olfactometry/mass spectrometry (GC-O/MS) assisted analysis. Heptanal, limonene, linalool, and trans-β-ionone were considered the fundamental odorants associated with the flavors of these teas. Comprehensive analysis showed that limonene, ethyl octanoate, copaene, ethyl butyrate (citrus), benzyl acetate, nerol (bergamot) and furfural (lemon) were determined as the characterized odorants for each type.

## 1. Introduction

Tea is now the second-most popular alcohol-free beverage worldwide and has great economic importance. Remarkably, so-called blend-tea (scented tea and flavored tea) maintains its popularity in Europe, the USA, and also nowadays, in the Asia zone for its unique flavor and remarkable health benefits. A wide variety of blend teas which combined different teas and various plant sources have been developed in the sale markets, such as citrus tea (citrus and black, green and white tea, orange flavor), jasmine green tea (dry jasmine flower and green tea, floral), blend lavender tea (lavender and black or green tea, herbal and floral), blend peach oolong tea (peach and oolong tea, sweet and fruity), and blend rose black tea (dry rose and black tea, floral) etc.

Among these blend teas, citrus black tea has a long history and remains popular among contemporary consumers [1]. For a long period, Chinese and European are accustomed to mixing black tea and citrus or its peel/oil seeking a fruit flavor tea. Nowadays, citrus species including *Citrus reticulate* (citrus), *Citrus limon* (lemon), *Citrus bergamia* (bergamot) were considered as the three main kinds of citrus blend black tea which we can get from the market [2,3,4,5]. However, the citrus black teas were difficult to be distinguished by ordinary consumers for the similar flavor characteristics and consistent packaging (tea bags), which provide easy opportunities for counterfeiters to adulterate the ungraded citrus sources. Therefore, the discrimination of the citrus black tea with different species was urgent and challenging.

The aroma is one of the important and determining factors of food. Hundreds of volatile components with different concentrations and odor characteristics comprise various types of aroma qualities of tea beverages [6,7]. As in most citrus fruits, the hydrocarbon monoterpene limonene (citrus-like, herbal odorant) contributes the most to the aroma quality of the corresponding citrus [8,9]. Notably, blend tea behaved more complex in flavor comparing with the traditional pure tea, for the superimposed and interaction effects among the enormous odorants sourced for teas and plant materials [10]. However, little is known about the aroma profiles and characterized odorants in commercial citrus blend black teas, although volatile compounds in different varieties of citrus, pure black, jasmine, and some traditional Earl Grey teas (bergamot/bergamot oil and black tea) have been investigated [11,12,13,14]. Interactions between odorants may be involved, but what impacts the unique flavor characteristics of teas blended with citrus remains obscure.

Advances in analytical technology have led to the development of comprehensive two-dimensional chromatography-time-of-flight mass spectrometry (GC×GC-TOFMS). This technology should be able to provide more comprehensive and precise chemical information about the aroma profiles of citrus black teas compared with a traditional gas chromatography-mass spectrometry [15]. However, although >5-fold more aromatic compounds could be separated by GC×GC-TOFMS, the main contributors to the overall scents of teas could not be determined by single chemical analysis, owing to diverse odor thresholds and concentrations of aromatic components. Thus, the active compounds had to be determined by evaluating the detailed odor characteristics of each individual aromatic component of the tea samples. We recently identified the key odorants in chestnut-like green teas and four of the most famous black teas using GC×GC-TOFMS combined with gas chromatography-olfactometry/mass spectrometry (GC-O/MS) [14,16]. This provided a technical reference for comparisons of characteristic odorants in citrus black teas.

Here, the study was aimed to differentiate and identify the aroma profiles and characterized odorants in citrus blend black tea with different citrus species by head space-solid phase microextraction (HS-SPME)/GC×GC-TOFMS combined with GC-O/MS technique. Our findings will significantly boost consumer understanding and distinguishing of emerging teas, offer a guide for producers seeking to improve and control the quality of citrus black teas.

## 2. Results

### 2.1. Sensory Evaluation

Fifty samples were divided into citrus, bergamot, and lemon groups (Appendix A) by main ingredient-different Citrus species. The bergamot group (BG) comprised black tea and bergamot peel or oil, which is popular in Europe called Earl Grey. The lemon group (LG) consisted of lemon peel with black tea, which is popular in China. The citrus group (CG) comprised the fruits of *Citrus reticulata* with black tea except for bergamot or lemon.

Sensory evaluation scored the representative samples as >4.0 (maximum = 8.0). Table 1 shows that 3 CG, 5 BG, and 4 LG samples had significant citrus-/lemon-like flavors. In CG, sweet, fruity and floral citrus-like were noted with a slight tea flavor, a bergamot-like and medical aroma with a slight tea flavor for BG, and a relatively heavy tea-like, clearly lemon-like and fresh flavor for LG.

### 2.2. Optimized Volatile Analyzing Approach Combining HS-SPME and GC×GC-TOFMS

As black tea composition is relatively large, the time of extraction is referred to as the 60 min extracted by Dr. Kang’s research [14]. Then the extraction fiber, tea-water proportion, and extraction temperature methods were optimized.

Among the investigated fibers, CAR/PDMS (57335-U) had the highest component numbers and total peak area (Appendix A: A_1_ and A_2_). The optimal proportion of tea to water was 1:4, which resulted in significantly higher component numbers and total peak area (Appendix A: B_1_ and B_2_). The number of compounds did not significantly differ among extraction temperatures, but the total peak area was significantly larger at 60 °C than others (Appendix A: C_1_ and C_2_).

Therefore, the optimized extraction conditions were CAR/PDMS fiber, tea-to-water ratio of 1:4, and extraction for 60 min at 60 °C.

### 2.3. Identification of Aroma Profiles in Citrus Blend Black Tea

About 800–1000 peaks were initially detected in samples with a minimum S/N ration of 50. After peak alignment, we initially identified 664 common volatile compounds by comparisons with mass spectra in the NIST 2014 library with a minimum similarity of 75%. Subsequently, the retention index (RI, the Kovats index) values of all compounds were calculated and compared with known RI to validate the accuracy of compound identification. The compounds whose RI value had a difference bigger than 20 by compared with the reported were deleted. Finally, 348 volatile compounds were ultimately identified as Reliable. These comprised 60 esters, 55 alkenes, 45 aldehydes, 45 ketones, 37 alcohols, 20 aromatic hydrocarbons, 20 oxyheterocyclic compounds, 19 nitrogen-containing compounds, 18 alkanes, 15 ethers, 7 phenols, 5 acids, 1 sulfur-compound, and 1 alkyne (Appendix A). Moreover, some important volatile compounds were verified by comparison with standards.

The distribution of the volatiles compounds in the citrus blend black teas is shown in Figure 1. The ratio (%) of alkenes was the highest (49.77% (CG); 47.16% (BG); 41.05% (LG)). Aldehyde was the second abundant compound classification with obvious differences existed in content levels among the three groups, that the contents in LG (26.31%) was twice higher than BG (11.20%). Ester was the third larger composition in CG (17.00%) and BG (16.82%); while ester (7.19%) was near with aromatic hydrocarbon (7.33%) in LG. The ranges of alcohol and aromatic hydrocarbon were 4.85–8.86% and 5.91–10.70%, respectively. The distribution of oxyheterocyclic compounds significantly differed among the three groups, being 6.24% in LG, but low in CG (0.22%) and BG (1.02%).

In detail, limonene was absolutely the highest constituent (CG, 35.41%; BG, 26.10%; LG, 29.44%), followed by β-myrcene (CG, 4.75%; BG, 5.73%; LG, 4.25%). Benzaldehyde was the main aldehyde, which was reported in many famous Chinese black teas, and considered as a key aroma compound in them contributing floral odorant [17,18]. *p*-Cymene, a common aromatic hydrocarbon in nature, with various biological activities, was detected in a high level (CG, 2.82%; BG, 6.93%; LG, 5.80%). Linalool, which is considered the most important odorant in most black teas [7,19], was also identified in blend teas at high ratios of 1.59–4.20%.

In addition to the above common higher-content components, different distribution of aroma profile was also exhibited among the three types. In CG, decanoic acid ethyl ester (6.95%), octanoic acid ethyl ester (4.74%), decanal (2.73%), *o*-cymene (1.94%) and ethyl butyrate (1.60%) took higher portions. α-terpineol (3.12%), β-pinene (1.58%), geranyl acetate (6.06%), *cis*-β-ocimene (4.90%), neryl acetate (2.93%), and sabinene (2.72%), were highly identified in BG. Neral (7.79%), furfural (4.23%), benzeneacetaldehyde (3.54%), 2-hexenal (2.94%), (E,E)-2,4-heptadienal (1.83%), geranyl acetate (1.75%), α-terpineol (1.74%), linalool (1.59%), and α-farnesene (1.40%) were identified in extracts at ratios >1% in LG.

### 2.4. Discrimination of Crucial Differential Volatiles in Citrus Blend Black Tea with Different Species

The aromatic profiles of three groups significantly differed. Therefore, we determined the key responsible volatiles that was considered as potentially characterized odorants of corresponding citrus blend black teas. We performed PLS-DA based on the normalized peak areas of 348 identified aroma compounds to obtain an overview of the distribution of differential volatile compounds among the three groups. The CG, BG, and LG were clearly discriminated by a PLS-DA model (R^2^Y = 0.907, Q^2^ = 0.836; Figure 2A), subsequent cross-validation confirmed the reliability of the model (R^2^ = 0.147, Q^2^ = −0.324; Figure 2B).

We screened 30 compounds as key differential volatiles based on variable importance of projection (VIP) values in the PLS-DA model with a threshold of 1.0 and *p* < 0.05 (Tukey s-b(K) tests). Subsequently, specific content differences of potentially characterized odorants were elucidated using hierarchical cluster analysis (HCA). The content distribution of key volatiles could be roughly divided into three classes (Figure 3). These classes comprised 12, 15, and 3 compounds that were obviously more abundant in CG, BG, and LG, respectively.

### 2.5. Identification of Aroma-Active Compounds in Citrus Blend Black Teas

Active aromatic compounds were assessed in equal amounts of mixed samples from each group using GC-O/MS. Table 2 shows that at least three panelists recognized 61 active aromatic compounds (Appendix A). By comprehensively combining the findings of panelists, we assigned 17 components to class A (fresh and green scents), 29 to class B (floral, fruity, or sweet scents), 9 to class C (herbal or woody scents), 2 to class D (bakery scents), and 4 to class E (unpleasant odor). The compounds in classes A and B might have directly contributed to the overall aroma quality of the citrus black teas owing to their similar scent types to sensory evaluation findings that all these samples have the obvious smell of fresh, fruity, sweet, and floral scents. Among them, heptanal (fresh, green; AI, 2.33–2.6), limonene (lemon-like, fruity, fresh; AI, 2.0–2.33), linalool (floral; AI, 2.57–3.29) and trans-β-ionone (floral; AI, 2.67–2.75) were detected in all groups.

The AI was notably highest for geranyl acetate (class B; AI, 3.29), β-pinene (class A; AI, 3.17), α-pinene (class B; AI, 3.17), linalool (class B; AI, 3.14), 3-(methylthio)-nonanal (class A; AI, 3.0), most of which were in class B. In addition to the concentrated distribution of odorants in class B, more compounds in LG belonged to classes D and E, which differed with the other groups.

GC-O/MS analysis identified 27, 23, and 29 active aromatic compounds in CG, BG, and LG, respectively. The compounds belonged to class A and B comprised a large proportion of the odorants in CG with moderate-to-high AI values (1.50–3.33). Among them, AI was the highest for 3-phenyl-2-propenal (class B; AI, 3.33), linalool (class B; AI, 3.29), β-myrcene (class A; AI, 3.17) and octanoic acid ethyl ester (class A; AI, 3.0) (>3.0). The 3-phenyl-2-propenal with a cinnamon-like aroma was considered as an exogenous compound that might a constituent of Orange & Cinnamon tea from Twinings (R. Twining and Co., Ltd., Andover, UK).

Similar to CG, most identified odorants belonged to class A and B in BG, geranyl acetate had the highest AI (2.86), followed by terpinen-4-ol (class A; AI, 2.83), trans-β-ionone (class B; AI, 2.75), β-pinene (class A; AI, 2.71), and 1-methyl-4-(1-methylethenyl)-benzene (class E; AI, 2.71).

A radar map based on the total AI values of odorants in each class was applied to determine overall flavor profiles and differences among the three types. Class B had the most outstanding flavor attributes, although their flavor profiles and corresponding AI significantly differed among all groups (Figure 4). The ANOVA results revealed that the most discriminative attributes were in class B (CG and BG, *p* < 0.01; BG and LG, *p* < 0.05), D (CG and LG, *p* < 0.01; BG and LG, *p* < 0.01), and E (CG and BG, *p* < 0.05; CG and LG, *p* < 0.01; BG and LG, *p* < 0.01), whereas class A and C did not significantly differ. The CG contributed most to the class B attribute, the total intensity (41.77) of the active aromatic compounds was significantly higher than other groups (BG, 24.70; LG, 36.52), which was basically consistent with the sensory evaluation. In addition to class B, the AI were significantly higher for class D and E in LG which may be responsible for the medical odor determined in the assessment of overall flavor quality in sensory evaluation. By contrast, all attributes in BG were less pronounced than other groups, but the scores of sensory evaluations did not significantly differ, indicating lower sensitivity of the sensory evaluation.

## 3. Discussion

In this study, the aroma profiles of different citrus blend black teas were investigated for the first time. Then, the aroma-active compounds were analyzed by GC-O/MS. Finally, a comprehensive conjoint analysis was made to identify the key aroma compounds in each group.

### 3.1. Aroma Profiles in Citrus Blend Black Tea

The ratio (%) of alkenes was the highest (49.77% (CG); 47.16% (BG); 41.05% (LG)) which agreed with previous findings of citrus [12]. Limonene and β-myrcene were the most important volatile components in orange (*Citrus sinensis*), lemon (*Citrus limon*), and mandarin (*Citrus reticulata*), and a very low content had also been identified in pure tea [12,13,19,20]. Compared with the previous report in lemon and tea, *p*-Cymene was much higher in our results [7,13,18,21]. It was supposed that *p*-cymene and linalool may come from both citrus and tea leaves. It might be the simple additive effects between the volatile compounds of pure tea and citrus.

The main volatile compounds identified in bergamot were also highly identified in BG [5,22]. These compounds might have been sourced from bergamot added during processing. A similar result was found in LG, in which LC and lemon had the same volatile compounds with high contents [2,13]. This might be due to the addition of citrus ingredients, resulting in a higher proportion of related citrus aroma profiles in the overall aroma composition of the blend tea. This made the citrus blend black tea contains a high content of alkenes and the resemble volatile components similar to the same cultivar citrus.

The obvious difference could be found in sensory evaluation results. The analysis of potentially characterized odorants of citrus blend black tea showed the key different volatile compounds of the three groups of blend tea samples, which might be the reason why the flavor was different among the three groups.

### 3.2. Aroma-Active Compounds in Citrus Blend Black Tea

In the result of GC-O/MS, the compounds belonged to class A and class B were considered as the basic odorants that contribute to the flavor of citrus blend black teas. Although the relative content of limonene was the highest, its aromatic intensity was notably moderate according to GC-O/MS, which may be due to its higher odor threshold [23]. Conversely, although the contents of heptanal (<1‰) and trans-β-ionone (<1‰) were far below than limonene, their extremely low odor thresholds of 3 μg/kg and 0.007 μg/kg, resulted in a similar AI to limonene [16]. The major source of linalool, which had low odor thresholds and high content in samples, was difficult to determine because it is a key volatile compound in both black tea and citrus [14,24].

Terpinen-4-ol, fresh aroma, has been detected both in bergamot and tea, which was consistent with our results [5,16]. Differently, linalool, which was reported as a key volatile in Earl grey black tea (bergamot black tea) [25], showed unobtrusive AI value (2.57) comparing with those in other groups, the difference of extraction methods might have caused this.

### 3.3. Comprehensive Understanding of Characterized Odorants in Citrus Blend Black Tea

The characterized odorants contributing to the aroma characteristics of citrus blend black teas were less rigorous when determined by simple quantitative or olfactory analyses due to the odor characteristics of different volatiles and subjective factors associated with panelists. Moreover, some of the GC×GC-TOFMS and GC-O/MS results did not always correspond, which was probably due to slight differences in the experimental conditions and error factors. Therefore, the combined results provided a more objective and precise identification of the characterized odorants in citrus blend black teas.

The distribution trends of the contents and AI values of seven compounds (CG (4), BG (2), LG (1)) were similar among the three groups, indicating their importance to the corresponding overall aroma quality (Figure 5). We considered that limonene (lemon-like, fruity, fresh), octanoic acid ethyl ester (green, waxy), copaene (sweet, floral), and ethyl butyrate (fruity) were the characterized odorants in CG. In fact, limonene was the most abundant and moderately intense in all samples, but the corresponding values in CG were significantly higher, indicating the superior distribution of the compounds in some citrus varieties. The other three odorants were essentially undetectable (0.39–3.64‰) in BG and LG. Their remarkably high distribution and scents may be responsible for the more intense floral and fruity scents in CG. Benzyl acetate (herbal, sweet) and nerol (floral, sweet) were determined as the characterized odorants in BG, which were not only detected in GC-O/MS but also showed more abundant. Nerol, which may be a unique volatile in BG, has previously been detected in Earl Grey tea, but its role has not been discussed in-depth, which may be due to differences in experimental design and detection methods [11,25]. Benzyl acetate might contribute to the unique herbal, medical, and sweet scents in BG. Similarly, furfural with a roasted odor was considered as one of the characterized odorants of LG.

## 4. Materials and Methods

### 4.1. Citrus Blend Black Tea Samples

A total of 12 representative citrus blend black tea samples including 3 CG samples, 5 BG samples and 4 LG samples were selected from 50 commercially available samples purchased at origin countries’ markets. Table 1 shows the brand names and blended ingredients. All samples were fully powdered (~200 mesh using a Tube Mill 100 control grinder (IKA Werke GmbH & Co. KG, Staufen, Germany) at 5000 rpm for 20 s.

### 4.2. Reagents and Materials

Aroma standards including neryl acetate, geranyl acetate, decanoic acid ethyl ester, coumarin, α-pinene, β-pinene, β-myrecene, γ-terpinene, α-terpinene, geraniol, limonene, copaene, hexanal, heptanal, benzaldehyde, octanal, (*E,E*)-2,4-heptadienal, benzeneacetaldehyde, decanal, α-ionone, linalool, terpinen-4-ol, nerol, o-cymene, 1-methyl-4-(1-methylethenyl)-benzene, 2-ethylfuran, furfural, theaspirane B, linalyl acetate, and trans-β-ionone, were purchased from J&K Scientific Ltd. (Beijing, China) and Sigma Aldrich Corp. (St. Louis, MO, USA). Distilled water was purchased from Wahaha Group Co. Ltd. (Hangzhou, China) and n-Alkanes (C8-C40) were obtained from J&K Scientific.

Headspace solid-phase microextraction (SPME) fibers including Carboxen^®^/polydimethylsiloxane (CAR/PDMS; 57335-U), polydimethylsiloxane/divinylbenzene (PDMS/DVB; 57327-U), divinylbenzene/carboxen/polydimethylsiloxane (DVB/CAR/PDMS; 57329-U), and polydimethylsiloxane (PDMS; 57301) were purchased from Supelco Inc. (Bellefonte, PA, USA).

### 4.3. Instrumentation and Equipment

Aroma constituents were analyzed using a Pegasus 4D GC×GC-TOF mass spectrometer (LECO Corp., St. Joseph, MI, USA). The first dimension (1-D) was a non-polar Rxi-5MS column (30 m × 250 μm × 0.25 μm) (Restek Corp., Bellefonte, PA, USA) and the second (2-D) was a moderate polar Rxi-17Sil MS column (1.9 m × 100 μm × 0.1 μm) (Restek Corp.) The GC-O analysis was conducted using a 7890B-5977B GC-MS system (Agilent Technologies Inc., Santa Clara, CA, USA) equipped with an ODP-3 Olfactory Detection Port (Gerstel GmbH & Co. KG, Mülheim an der Ruhr, Germany).

### 4.4. Optimization of Volatile Extraction from Citrus Blend Black Teas Using HS-SPME (for GC×GC-TOFMS and GC-O/MS Analyses)

Using a multiple-factor orthogonal experiment to determine the appropriate extraction fibers among CAR/PDMS (57335-U, 85 μm), PDMS/DVB (57327-U, 65 μm), DVB/CAR/PDMS (57329-U, 50/30 μm) and PDMS (57301, 100 μm), the experiment with water-sample ratio and temperature proceeded as follows. The fibers were conditioned at high temperature (300 °C for CAR/PDMS, 250 °C for PDMS/DVB and PDMS, 270 °C for DVB/CAR/PDMS) for 0.5 h before their first use and then screened under the same HS-SPME conditions. Triplicate powdered samples (1.0 g) were each placed in 20-mL glass vials, then boiling water (2, 3, 4, 6, or 8 mL) was added. The vials were immediately placed in a heating oscillator to equilibrate for 3.0 min at 30 °C, 50 °C, 60 °C, 70 °C, 90 °C, respectively, then the solid-phase microextracted fibers (SPME) were exposed to the vial headspace, and stirred at a constant speed and temperature for 60 min. Finally, the SPME fibers were loaded into the GC×GC injector and left for 5.0 min to permit thermal desorption of the aroma extract.

### 4.5. GC×GC-TOFMS Analysis

GC×GC conditions: The temperature of the GC injector and the transfer line was set to 250 °C. Helium (99.999%) was the carrier gas at a constant flow of 1.0 mL/min. A split injection was applied at a split ratio of 20:1. Standards in ethanol were injected using an MPS-2 multi-purpose sampler with an injection volume of 1.0 μL, and the aroma extracts were injected using the HS-SPME auto sampling system. The temperature programs were as follows: hold at 50 °C for 2 min, increase in 8 °C/min increments to 265 °C, then hold for 5.0 min for the 1-d column, and hold at 55 °C for 2 min, increase in 8 °C/min increments to 270 °C, the hold for 5 min for the 2-d column. The modulation period was set at 5 s. TOFMS conditions: The TOFMS parameters were electron ionization at −70 eV, an ion source temperature of 220 °C, an electron multiplier at 1400 V and a mass range of 33–600 u.

### 4.6. GC-O/MS Analysis

GC-MS conditions: HP-5MS column, 30 m × 250 μm × 0.25 μm; GC injector temperature, 250 °C; helium (99.999%) flow, 1.6 mL/min; splitless injection and 0.0 s of modulation. The temperature program for the GC column proceeded as follows: hold at 50 °C for 3 min, increase in 4 °C/min increments to 265 °C, and then hold for 5 min. The temperature of the transfer line was 270 °C for the entire 60.75-min duration of the analysis. Mass spectrometry proceeded under an ion source temperature of 220 °C with a mass range of 33–600 u, detector voltage, 1300 V and electron ionization −70 eV.

Three male and four female panelists who were selected and trained as we have previously described, tested the aroma-active compounds in each sample three times using GC-O [26]. The intensity of each aroma was defined on a scale of 1 to 4 as weak (1), moderate (2), strong (3), and extremely strong (4) [16,27]. A general description of odorants with the same retention time by at least three panelists was selected and further determined by GC-MS and standards. The panelist scores were averaged to define the corresponding aroma intensity.

### 4.7. Sensory Evaluation

Two male and three female healthy tea-tasters conducted the sensory evaluation. All were selected and trained as we have previously described [14]. They had been certified by the China Tea Science Society after passing theory and practical examinations and had at least two years of experience in tea sensory evaluation.

The aroma characteristics of the samples were described according to the national standards in “Black tea” (GB/T 13738-2017) and “Teabag” (GB/T 24690-2018). The aroma qualities of samples were separated into two parts. The tea-tasters should score the intensity of the tea and citrus aromas that were smelled in each sample, respectively. Scores were based on the aroma intensity (AI) method in GC-O with the modification of adding a score of 1 to 4 indicate the absence of a tea or citrus fragrance, and a score of 0 meant that there was no such scent. The final score was the sum of the averaged dimensional intensities determined by the tasters.

### 4.8. Data Processing

The GC×GC-TOFMS data preprocessing method in the LECO Chroma TOF software was used. Partial least squares discriminant analysis (PLS-DA) using the SIMCA-P 12.0 software (Umetrics Corporation, Umeå, Sweden) and hierarchical clustering analysis (HCA) using the MultiExperiment Viewer 4.8.1 (Oracle Corporation, Redwood Shores, California CA, USA) were performed to the statistical multivariate analyses. ANOVA analysis was performed using SPSS Statistics 20.0 (IBM Corp., Armonk, NY, USA).

## 5. Conclusions

In conclusion, the aroma profile and characteristic odorants in the main current commercial citrus blend black teas were thoroughly investigated utilizing HS-SPME-GC×GC-TOFMS combined with GC-O/MS techniques for the first time. A total of 348 volatile compounds were ultimately identified that consisted of >50% alkenes and aldehydes. The PLS-DA resulted in 30 significant differential volatile compounds among the three types. Moreover, GC-O/MS analysis revealed 61 aroma-active compounds. Most of these compounds presented green, fresh, floral, lemon-like fruity, and sweet scents. Especially heptanal, limonene, linalool, and trans-β-ionone were considered as the basic odorants for citrus blend black tea flavors. The combination of GC×GC-TOFMS and GC-O/MS indicated that each group had a unique composition of volatile compounds, namely, limonene, octanoic acid ethyl ester, copaene, ethyl butyrate in CG, benzyl acetate, and nerol in BG, and furfural in LG. The combination of results reliably identified blended tea aromas and flavors and led to a comprehensive understanding of the flavor sources in citrus blend black tea. Meaningful, the result also lays the foundation for the cultivar discrimination and aroma quality control of the popular blended teas. Our subsequent studies will focus on identifying unknown odorants, improving analytical approaches, and uncovering synergistic and inhibitory effects among odorants.

## Figures and Tables

**Figure 1 molecules-25-04208-f001:**
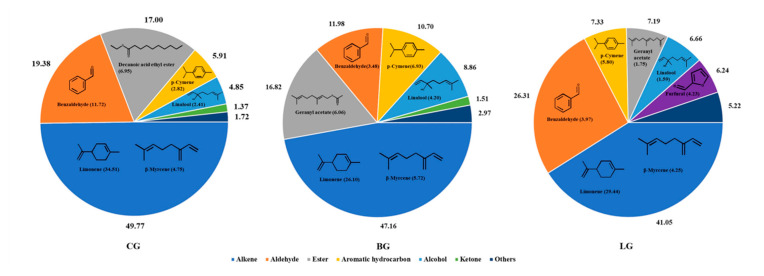
Ratios (%) of volatile compounds in three groups of citrus blend black teas. **CG** (Citrus group), others: ether (0.94), alkane (0.25), oxyheterocyclic compound (0.22), acid (0.12), nitrogen-containing compound (0.11), phenol (0.08), sulfocompound (0.004); **BG** (Bergamot group), others: ether (1.25), oxyheterocyclic compound (1.06), phenol (0.25), nitrogen-containing compound (0.20), alkane (0.12), acid (0.07), sulfocompound (0.004), alkyne (0.003); **LG** (Lemon group), others: ketone (3.05), ether (0.85), acid (0.52), nitrogen-containing compound (0.50), alkane (0.20), phenol (0.10), sulfocompound (0.001), alkyne (0.001).

**Figure 2 molecules-25-04208-f002:**
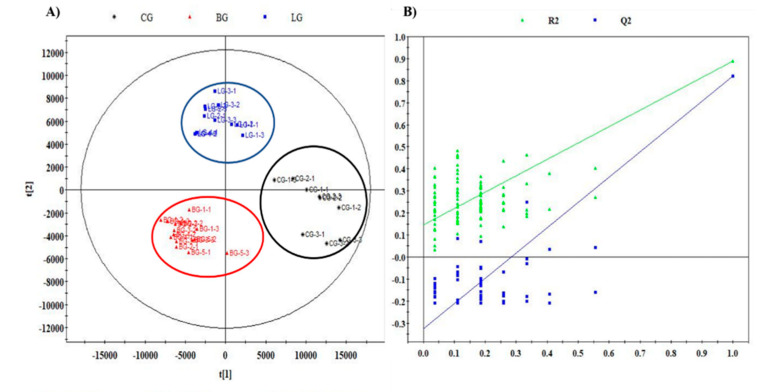
The PLS-DA plot and cross-validation of the three citrus blend black tea groups. (**A**) PLS-DA plot (R^2^Y = 0.907, Q^2^ = 0.836); (**B**) Cross-validation of PLS-DA model with 100 permutation tests (R^2^ = 0.147, Q^2^ = −0.324).

**Figure 3 molecules-25-04208-f003:**
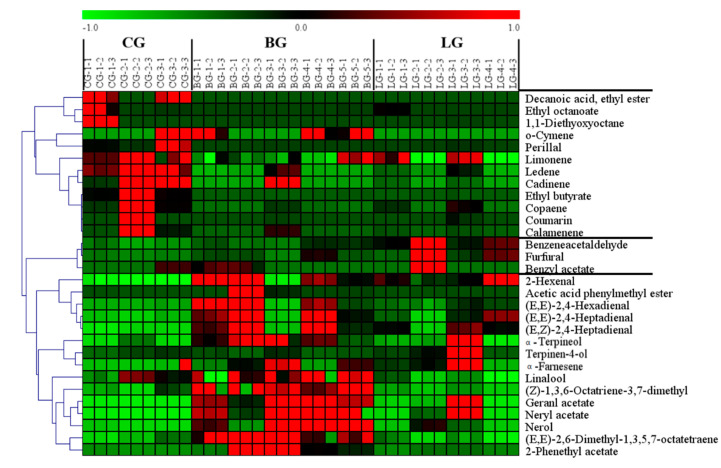
Heat map of contents of key differential volatile compounds among three groups of citrus blend black tea.

**Figure 4 molecules-25-04208-f004:**
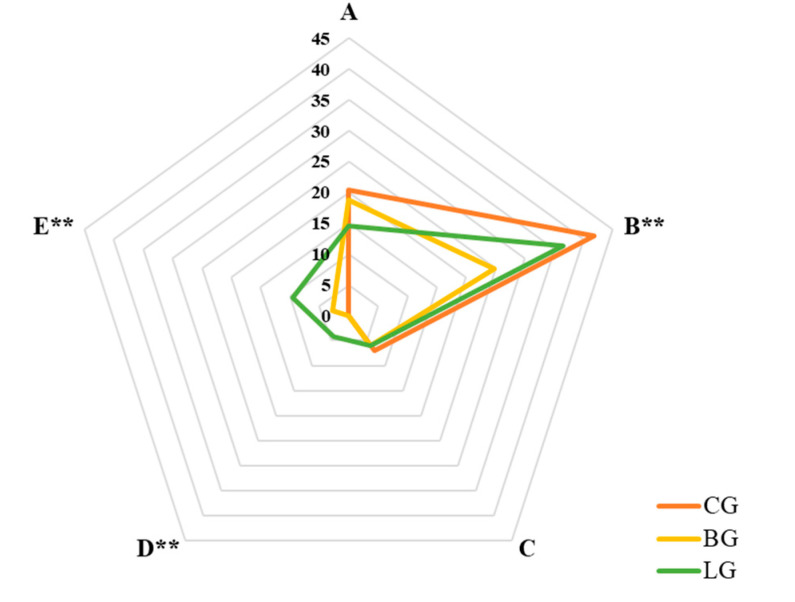
Distribution of five classes of 61 active aromatic compounds according to nature of their scents in citrus blend black teas. * Significant differences among three groups (**, *p* < 0.01;). Classes: A, fresh and green; B, floral, fruity, and sweet; C, herbal and woody; D bakery; E, unpleasant.

**Figure 5 molecules-25-04208-f005:**
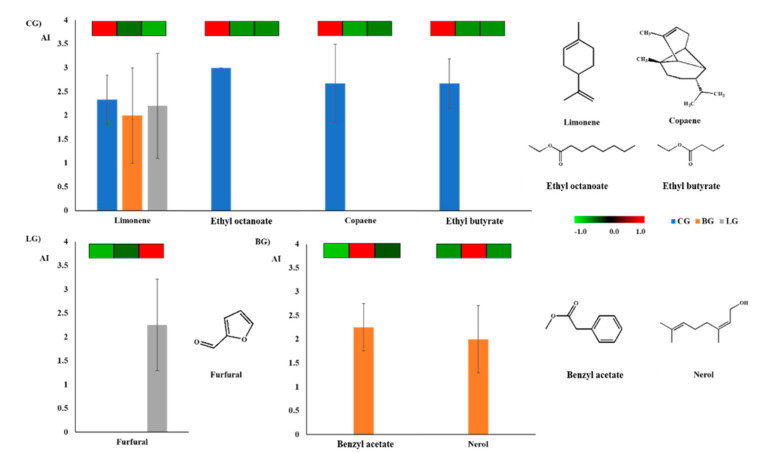
The GC-O/MS aroma intensity and abundance heat map in GC×GC-TOFMS analysis of key aroma compounds in three citrus blend black tea groups. CG: Citrus Group; BG: Bergamot Group; LG: Lemon Group.

**Table 1 molecules-25-04208-t001:** Information and sensory evaluation of 12 typical citrus blend black teas.

Group	No.	Brand	Origin	Ingredient	Score	Description
Citrus group (CG)	CG-1	LUPICIA-Iyo no Kaori	Japan	Black tea, Iyo citrus peel	4.90	Slightly tea flavor, citrus-like, fruity, sweet
	CG-2	TWININGS-Orange & Cinnamon Tea	Poland	Black tea, cinnamon peel 20%, citrus flavor 6%, citrus slice 1%	4.20	Slightly tea flavor, cinnamon-like, citrus-like, fruity, floral, herbal medical
	CG-3	LUPICIA-Karakoro	Japan	Black tea, grapefruit, fried rice, sugar	4.00	Slightly tea flavor, citrus-like, sweet, floral, fruity
Bergamot group (BG)	BG-1	TEEKANNE-Earl Grey	Germany	Black tea, bergamot oil	4.10	Slightly tea flavor, sweet, citrus-like, bergamot oil-like
	BG-2	TEEKANNE-Earl Grey Jasmine	Germany	Black tea, bergamot spice, jasmine spice	4.10	Slightly tea flavor, bergamot-like, jasmine-like, floral, fresh, grassy
	BG-3	HEME-Earl Grey Lavender	United Kingdom	Black tea, bergamot oil flavoring, dried marigold petals, dried lavender buds	4.14	Slightly tea flavor, lavender-like, bergamot-like, medical
	BG-4	TWININGS-Earl Grey	Poland	Black tea, bergamot oil	4.00	Slightly tea flavor, wood, medical, fruity, bergamot-like
	BG-5	LUPICIA-Eary Grey	Japan	Black tea, bergamot oil	4.44	Slightly tea flavor, bergamot-like, fruity, slightly medical
Lemon group (LG)	LG-1	TWININGS-Lemon Scented Tea	Poland	Black tea, lemon	4.10	Heavy tea flavor, sour lemon-like, fresh
	LG-2	MeeCoo-Lemon Black Tea	China	Congfu black tea, lemon	4.40	Heavy tea flavor, fresh, lemon-like, roasted, sweet
	LG-3	Lipton-Lemon Black Tea	China	Black tea, lemon peel	4.04	Havey tea flavor, fresh, lemon-like, caramel-like
	LG-4	TEEKANNE-Fresh Lemon	Germany	Black tea, lemon concentrate (19%), lemon	4.40	Heavy tea flavor, fresh, lemon-like, flavor, fruity, roasted, sweet

**Table 2 molecules-25-04208-t002:** Active aromatic compounds identified in citrus blend black tea.

Class ^[1]^	No.	Compounds	Aroma Intensity			Ordor Characteristic
			CG	BG	LG	
A	1	Hexanal *		2.00	2.00	Fresh
	2	Heptanal *	2.60	2.57	2.33	Fresh, Green
	3	β-Myrcene *	3.17	2.00		Green, Metallic
	4	β-Pinene *		2.71	3.17	Green, wood
	5	Carveol *			2.00	Fresh
	6	(*E,E*)-2,6-Dimethyl-2,4,6-octatriene		2.25		Fresh, Floral
	7	Citronellal	2.50			Green, Wood
	8	3,6-Dihydro-4-methyl-2-(2-methyl-1-propenyl)-2*H*-pyran		2.00		Green
	9	3-(Methylthio)-nonanal			3.00	Green, Wood
	10	Terpinen-4-ol *		2.83		Fresh, Wood
	11	Verbenol			2.00	Fresh, Herbal
	12	Ethyl octanoate	3.00			Green, Waxy
	13	2-(*n*-Propyl)-pyrazine	1.50			Green, Limon-like
	14	4-(1-Methylethyl)-benzaldehyde	2.71			Fresh, Herbal
	15	Bornyl acetate	2.33			Fresh, Wood
	16	Decanoic acid ethyl ester		2.33		Green, Fatty
	17	Dodecanal	2.60			Green, Waxy
B	18	Ethyl butyrate	2.67			Fruity
	19	(*E*)-2-Hexenal *	2.43	1.33		Fruity
	20	Octanal *	2.67			Lemon-like, Fresh
	21	Nerol *		2.00		Floral, Sweet
	22	4,6,6-Trimethylbicyclo[3.1.1]hept-3-en-2-one		2.17		Fruity
	23	Limonene *	2.33	2.00	2.20	Lemon-like, Fruity, Fresh
	24	Benzeneacetaldehyde *	2.60		2.25	Floral
	25	α-Pinene *			3.17	Floral
	26	Linalool *	3.29	2.57	3.14	Floral
	27	*p*-Mentha-1,8-dien-7-ol	2.17		2.20	Floral, Green
	28	α-Terpineol *		2.00	2.33	Floral
	29	Decanal	2.83			Sweet
	30	(*Z*)-3,7-Dimethyl-2,6-octadienal			2.29	Lemon-like, Fresh
	31	Geraniol *			2.86	Lemon-like, Fresh
	32	Linalyl acetate *		2.57		Citrus-like, Herbal
	33	(*E*)-3,7-Dimethyl-2,6-Octadienal			2.40	Lemon-like
	34	3-Phenyl-2-propenal	3.33			Sweet, Wood, Cinnamon-like
	35	Citral			2.67	Lemon-like
	36	Neryl acetate *	1.80		2.57	Floral
	37	a-Copaene	2.67			Sweet, Floral
	38	Geranyl acetate *		2.86	3.29	Floral, Sweet
	39	β-Caryophyllene	2.75			Floral
	40	β-Cubebene	2.40			Fruity, Citrus-like
	41	Jasmine lactone			2.40	Floral
	42	α-Ionone *		2.25		Sweet, Floral
	43	γ-Decalactone	2.83			Floral
	44	Nerolidol		2.20		Fruity
	45	trans-β-Ionone *	2.67	2.75	2.75	Floral
	46	a-Calacorene	2.33			Floral
C	47	γ-Terpinene *			2.00	Herbal, Green
	48	1-Ethenyl-4-methoxybenzene			2.00	Wood
	49	Benzyl acetate		2.25		Herbal, Sweet
	50	Isopulegol acetate		2.33		Wood, Sweet
	51	2,6,10,10-Tetramethyl-1-oxaspiro[4.5]dec-6-ene *			2.00	Herbal
	52	Aromandendrene	2.86			Wood, Sweet
	53	(*E*)-β-Famesene	2.50			Wood, Sweet
	54	α-Muurolene	1.67			Wood
	55	Caryophyllene oxide		1.33		Herbal, Sweet
D	56	Furfural *			2.25	Roasted
	57	Salicylic acid			2.00	Roasted
E	58	2-Ethylfuran *			2.00	Unpleasant, Medical
	59	Benzaldehyde *			2.20	Unpleasant, Medical
	60	1-Methyl-4-(1-methylethenyl)-benzene *		2.71	2.67	Unpleasant, Wood
	61	Isopulegol			2.71	Unpleasant, Wood

Note: *: the compound was identified by authentic standards; ^[1]^ the classification of odor characteristics of each compounds, **Class A**: fresh and green scents; **Class B**: floral, fruity and sweet scents; **Class C:** herbal and wood scents; **Class D**: a bake scent; **Class E**: an unpleasant scent.

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
