# Peer review of "Discrimination and Identification of Aroma Profiles and Characterized Odorants in Citrus Blend Black Tea with Different Citrus Species"

_molecules, 2020, doi:10.3390/molecules25184208_

Round 1
Reviewer 1 Report
Line 120: Only three groups? Figure 1 shows more than 3 groups of compounds. Or the groups are referred to the citrus blend black teas. If so, please reconsider the term "groups". Suggestion: The distribution of the volatiles compounds in the citrus blend black teas is shown in Figure 1.
Line 126: Add "respectively" after "10.70%".
LIne 147: Add a space and start a new phrase after "BG". Neral...
Line 150. Delete the whole phrase after "LG". 3.4 Discrimination ...
Line 233: "contended"?. Verify.
Line 254: Suggestion: groups. The difference of extraction methods might have caused this.
Line 270: ... BG and LG. Their remarkably high ...
Line 366: ... sweet scents. Especially heptanal, ...
Line 369: ... ethyl butyrate in CG, benzyl acetate, ...
Author Response
Response to Reviewer 1 Comments
Point 1: Line 120: Only three groups? Figure 1 shows more than 3 groups of compounds. Or the groups are referred to the citrus blend black teas. If so, please reconsider the term "groups". Suggestion: The distribution of the volatiles compounds in the citrus blend black teas is shown in Figure 1. 

Response 1: Line 121: the groups are referred to the classification of citrus blend black teas. The first sentence in line 121 have been revised into: The distribution of the volatiles compounds in the citrus blend black teas is shown in Figure 1 according to the reviewer’s suggestion (lines 121-122 of page 4).
Point 2: Line 126: Add "respectively" after "10.70%".
Response 2: The "respectively" has been added after "10.70%", please check (line 127-128 of page 4).
Point 3: Line 147: Add a space and start a new phrase after "BG". Neral...
Response 3: The space has been added after "BG". And the next sentence started with "Neral…" (line 149 of page 5).
Point 4: Line 150. Delete the whole phrase after "LG". 3.4 Discrimination ...
Response 4: Line 152: the last sentence had been deleted (lines 152-153 of page 5).
Point 5: Line 233: "contended"?. Verify.
Response 5: The sentence has been revised into "A similar result was found in LG, in which LC and lemon had the same volatile compounds with high contents [2, 13]" (lines 235-236 of page 9).
Point 6: Line 254: Suggestion: groups. The difference of extraction methods might have caused this.
Response 6: The sentence has been revised according to the reviewer’s suggestion (line 256 of page 10).
Point 7: Line 270: ... BG and LG. Their remarkably high ...
Response 7: The sentence has been divided into two sentences: "The other three odorants were essentially undetectable (0.39‰â€’3.64‰) in BG and LG. Their remarkably high distribution and scents may be responsible for the more intense floral and fruity scents in CG" (lines 272 of page 10).
Point 8: Line 366: ... sweet scents. Especially heptanal, ...
Response 8: The sentence has been divided into two sentences: “Most of these compounds presented green, fresh, floral, lemon-like fruity, and sweet scents. Especially heptanal, limonene, linalool, and trans-β-ionone were considered as the basic odorants for citrus blend black tea flavors.” (lines 372 of page 13).
Point 9: Line 369: ... ethyl butyrate in CG, benzyl acetate, ...
Response 9: The capitalization error has been corrected (line 375 of page 13).
We acknowledge Reviewer 1 for his/her cautious and conscientious work!
Reviewer 2 Report
The manuscript under appreciation is about the characterization of the aroma profiles of citrus blend black tea with different citrus species using HS-SPME- GC×GC-TOFMS and GC-O/MS. A classification and discrimination of samples using HCA and PLS-DA, was also performed.
The manuscript is interesting and provides novelty and the results are important for future application.
The following comments are to be taken into account by the authors:
- Lines 263-277: The determination of key volatiles for the discrimination of samples is very important. The authors report the characteristic odorants of citrus, bergamot and lemon blend black tea based on AI. Are these compounds included in the 30 volatiles identified based on VIP values (line 160)? Please discuss.
- Table 1. No. LG-3 under Description, please correct Have tea flavor to Heavy tea flavor.
- Subsection 4.1: Please report the number of samples for each of the three citrus blend black tea groups.
- Line 113: Please clarify how you determined the RI values (Kovats index or arithmetic index?).
- Please report if the SPME fibers were conditioned before their first use.
- Please provide in the Supporting Information an indicative GCxGC chromatogram from citrus, bergamot and lemon blend black tea.
- Line 312: Please report the purity grade of water.
- Subsection 4.8: Please clarify which software was used for ANOVA.
- Table S2: After line 60, correct alkenens to alkenes. Line 42 under the table S2, correct The compounds was identified… to The compounds were identified.
Author Response
Response to Reviewer 2 Comments
Point 1: Lines 263-277: The determination of key volatiles for the discrimination of samples is very important. The authors report the characteristic odorants of citrus, bergamot and lemon blend black tea based on AI. Are these compounds included in the 30 volatiles identified based on VIP values (line 160)? Please discuss.

Response 1: Yes, the characteristic odorants in the citrus blend black tea with different citrus species were determined by comprehensive analysis of GC-O/MS, GC×GC-TOFMS and multivariate statistical analysis results. The common compounds in Fig. 3 (the key differential volatiles with VIP>1) and Table 2 (aroma-active compounds) were considered as the potential characteristic odorants of citrus, bergamot and lemon blend black tea, and the above odorants with the similar distribution trends between contents and AI values were ultimately identified as the characteristic odorants.
In this study, ethyl octanoate, ethyl butyrate, limonene, and copaene which presented significant higher content levels and AI values in CG than those in LG and BG were determined as the characteristic odorants in citrus blend black tea. Similarly, benzyl acetate & nerol and furfural were determined as characteristic odorants in lemon and bergamot blend black teas, respectively.
Point 2: Table 1. No. LG-3 under Description, please correct Have tea flavor to Heavy tea flavor.
Response 2: The spelling mistake has been corrected. (“No. LG-3 under Description” in Table 1 of page 3).
Point 3: Subsection 4.1: Please report the number of samples for each of the three citrus blend black tea groups.
Response 3: The number of samples for each of the three citrus blend black tea groups has been added in subsection 4.1 (lines 285-286 of page 11).
Point 4: Line 113: Please clarify how you determined the RI values (Kovats index or arithmetic index?).
Response 4: In this study, Kovats index was used to determine the RI values. And the related description has been added in the manuscript (lines 112-113 of page 4).
Point 5: Please report if the SPME fibers were conditioned before their first use.
Response 5: All SPME fibers were conditioned before their first use in this study, and the related description on the film thickness, conditioning temperature and time of each fiber has been added in the manuscript (lines 312-315 of page 12).
Point 6: Please provide in the Supporting Information an indicative GCxGC chromatogram from citrus, bergamot and lemon blend black tea.
Response 6: The representative GC×GC chromatograms of tea aroma extracts in each group were provided in the Supporting Information as Figure S2, Figure S3, and Figure S4. (Line 9-14 of page 1-2 in the Supporting Information).
Point 7: Line 312: Please report the purity grade of water.
Response 7: The water used in this study was distilled water purchased from Wahaha Group Co. Ltd. (Hangzhou, China) (line 296 of page 11).
Point 8: Subsection 4.8: Please clarify which software was used for ANOVA.
Response 8: ANOVA analysis was performed using SPSS Statistics 20.0 (IBM Corp., Armonk, NY, USA) (lines 363-364 of page 13).
Point 9: Table S2: After line 60, correct alkenens to alkenes. Line 42 under the table S2, correct The compounds was identified… to The compounds were identified.
Response 9: The spelling mistakes have been corrected (after No. 60 in Table S2 and line 47 of Supporting Information).
We acknowledge Reviewer 2 for his/her cautious and conscientious work!
Round 2
Reviewer 2 Report
The authors have adressed all issues except one. Please check again the correction at Table 1. No. LG-3 under Description, have tea flavor has been wrongly corrected to havey tea flavor, instead of heavy.
If the above be corrected, the manuscript is ready for publication.